# Antitumor Effect of *Glandora rosmarinifolia* (Boraginaceae) Essential Oil through Inhibition of the Activity of the Topo II Enzyme in Acute Myeloid Leukemia

**DOI:** 10.3390/molecules27134203

**Published:** 2022-06-29

**Authors:** Manuela Labbozzetta, Paola Poma, Chiara Occhipinti, Maurizio Sajeva, Monica Notarbartolo

**Affiliations:** Department of Biological, Chemical and Pharmaceutical Science and Technology (STEBICEF), University of Palermo, 90128 Palermo, Italy; chiara.occhipinti17@gmail.com (C.O.); maurizio.sajeva@unipa.it (M.S.); monica.notarbartolo@unipa.it (M.N.)

**Keywords:** EOs, napthoquinone, Topo II, multidrug resistance

## Abstract

It was previously shown that the antitumor and cytotoxic activity of the essential oil (EO) extracted from the aerial parts of *Glandora rosmarinifolia* appears to involve a pro-oxidant mechanism in hepatocellular carcinoma (HCC) and in triple-negative breast cancer (TNBC) cell lines. Its most abundant compound is a hydroxy-methyl-naphthoquinone isomer. Important pharmacological activities, such as antitumor, antibacterial, antifungal, antiviral and antiparasitic activities, are attributed to naphthoquinones, probably due to their pro-oxidant or electrophilic potential; for some naphthoquinones, a mechanism of action of topoisomerase inhibition has been reported, in which they appear to act both as catalytic inhibitors and as topoisomerase II poisons. Our aim was to evaluate the cytotoxic activity of the essential oil on an acute myeloid leukemia cell line HL-60 and on its multidrug-resistant (MDR) variant HL-60R and verify its ability to interfere with topoisomerase II activity. MTS assay showed that *G. rosmarinifolia* EO induced a decrease in tumor cell viability equivalent in the two cell lines; this antitumor effect could depend on the pro-oxidant activity of EO in both cell lines. Furthermore, *G. rosmarinifolia* EO reduced the activity of Topo II in the nuclear extracts of HL-60 and HL-60R cells, as inferred from the inability to convert the kinetoplast DNA into the decatenated form and then not inducing linear kDNA. Confirming this result, flow cytometric analysis proved that EO induced a G_0_-G_1_ phase arrest, with cell reduction in the S-phase. In addition, the combination of EO with etoposide showed a good potentiation effect in terms of cytotoxicity in both cell lines. Our results highlight the antitumor activity of EO in the HL-60 cell line and its MDR variant with a peculiar mechanism as a Topo II modulator. Unlike etoposide, EO does not cause stabilization of a covalent Topo II-DNA intermediate but acts as a catalytic inhibitor. These data make *G. rosmarinifolia* EO a potential anticancer drug candidate due to its cytotoxic action, which is not affected by multidrug resistance.

## 1. Introduction

The plant kingdom represents one of the most important sources of active principles in which numerous biological activities have been recognized. In particular, EOs play important functions both in plant life, such as protection against plants or vermin, repellent or recall action for pollinating insects and as raw materials for the synthesis of new drugs [1]. The biological properties of EOs have been demonstrated in the treatment of several diseases, as well as being effective in increasing the bioavailability of other drugs. Their in vitro anticancer, anti-bioceptive, antiviral, antiphlogistic and antimicrobial activities are widely documented in the vast literature [2,3,4,5,6]. EOs are highly fat soluble and are characterized by low toxicity and often by a multi-target activity, precisely because they are composed of several molecules with different pharmacological actions. Our previous studies have shown that EOs from plants have antitumor activities in different types of tumors, such as HCC, TNBC [7,8] and acute myeloid leukemia (AML) [9], all characterized by low responsiveness to chemotherapeutic drugs.

EO extracted from the aerial parts (branches with leaves) of *Glandora rosmarinifolia* (Ten.) D.C. Thomas (Boraginaceae) has previously shown antitumor and cytotoxic activities involving a pro-oxidant mechanism in HCC and TNBC cell lines [7]. The chemical composition of *G. rosmarinifolia* EO has been identified [7], and an isomer of hydroxy-methyl-naphthoquinone was among the most abundant compounds (5.3%).

Naphthoquinones are quinones with another aromatic ring fused onto them and usually occur as glycosides. Naphthoquinones are not widespread but are found in several plant families (e.g., Bignoniaceae, Ebenaceaeos, Droseraceae, Juglandaceae, Plumbaginaceae, Boraginaceae, among others). Interestingly, naphthoquinones are also biosynthesized in various algae, fungi, some animals and also as metabolic products in some bacteria [10]. Important pharmacological activities, such as antitumor, antibacterial, antifungal, antiviral and antiparasitic activities, are attributed to them [11,12,13,14]. Naphthoquinones can act mainly as pro-oxidants, reducing oxygen to reactive oxygen species (ROS) or as electrophiles, forming covalent bonds with tissue nucleophiles [15,16]. The extensively studied naphthoquinones [17,18,19,20,21,22,23,24,25,26,27,28,29,30] showed anticancer activity both in vitro and in vivo via many molecular mechanisms, such as targeting apoptosis, autophagy pathway, cell cycle arrest, anti-angiogenesis pathway, anti-invasion and anti-metastasis pathway. Some of these possess potent antitumor activity by inhibiting the activity of DNA topoisomerase I/II [31,32,33,34].

Topoisomerases (Topo I and II) are important enzymes acting on the topology of DNA during its replication and transcription; they both regulate the winding of DNA. Topo I cuts one strand of DNA, while Topo II cuts both strands of DNA double helix to relax it. Topo II, an essential nuclear enzyme that regulates the topology of DNA, is a cellular target for several clinically important anticancer agents, such as anthracyclines (adriamycin, doxorubicin), epipodophyllotoxins (etoposide, teniposide), anthracenedione (mitoxantrone) and aminoacridines (m-AMSA) [35,36]. Topo II inhibitors are classified into two groups according to their inhibition mechanism. The first group, termed Topo II poisons, stabilize the cleavable complex by preventing the religation step, thereby generating double-stranded DNA breaks. The second group, referred to as catalytic inhibitors, block Topo II at specific sites in its catalytic cycle and interfere with its binding to the DNA. Topo-II inhibitors generally induce S-phase arrest [37,38].

In the present study, we wanted to evaluate the antitumor activity of *G. rosmarinifolia* EO in an acute myeloid leukemia cell line HL-60 and its variant, HL-60R, characterized by scarce responsiveness to chemotherapeutic drugs with an MDR phenotype. The presence of hydroxyl-methyl-naphthoquinones among the major components of EO led us to investigate the ability of the oil to interfere with the activity of topoisomerase II. We observed as *G. rosmarinifolia* EO reduced topoisomerase II activity, as inferred by its inability to convert kinetoplast DNA to the decatenated form, highlighting a further antitumor action mechanism of the EO, in addition to the pro-oxidant one.

## 2. Results and Discussion

### 2.1. In Vitro Anticancer Activity of G. rosmarinifolia Essential Oil

We examined the cytotoxic activity of *G. rosmarinifolia* EO on the HL-60 cell line and on its MDR variant HL-60R using the 3-(4,5-dimethylthiazol-2-yl)-5-(3-carboxymethoxyphenyl)-2-(4-sulphophenyl)-2Htetrazolium (MTS) assay. As shown in Figure 1, after 72 h of treatment at different concentrations (10–50 µg/mL), the EO induced inhibition of cell growth in a concentration-dependent and equivalent manner in both cell lines (A and B). The EO did not show cytotoxic activity at the same concentrations on the non-tumor cell line hTERT RPE-1 (C).

Table 1 reports the IC_50_ and the resistance factors (RF) of *G. rosmarinifolia* EO and reference drugs in the cell lines. The IC_50_ values of EO are, respectively, 36.75 ± 0.2 µg/mL in the HL-60 cell line and 37 ± 0.7 µg/mL in the HL-60R cell line. The resistance factor of EO on MDR cells was 1.0, much lower than that of the reference drug etoposide (RF 148.0). The variant HL-60R was obtained by treating HL-60 cells with increasing doses of doxorubicin, and its molecular characterization was carried out previously [39]. Treatment with doxorubicin modified the phenotype of the HL-60 cell line by inducing the expression of several factors responsible for multidrug resistance. The HL-60R cell line was characterized by the overexpression of P-glycoprotein (P-gp) efflux pump, constitutive activation of the nuclear factor kappa B (NF-κB) and the consequent overexpression of the inhibitor of apoptosis proteins (IAPs). These data led us to the hypothesis that the EO was able to overcome multidrug resistance and for this reason, we assume that EO was not a substrate of P-gp as doxorubicin and etoposide. To verify this hypothesis, we performed the cytotoxic assay with the combination of EO and verapamil, a P-gp inhibitor; we did not observe any potentiation or synergistic effects, confirming that the EO was not a substrate of the P-gp efflux pump (data not shown). 

### 2.2. In Vitro Pro-Oxidant Activity of G. rosmarinifolia Essential Oil

In a previous study, Poma et al. [7] suggested that the antitumor and cytotoxic activity shown by the EO of *G. rosmarinifolia* in HCC and TNBC cell lines involves a pro-oxidant mechanism through an increase in free radical generation due to the presence of diterpenes and hydroxy-methyl-naphthoquinone. Therefore, we wanted to investigate whether the antitumor activity of *G. rosmarinifolia* EO was also due to the pro-oxidant activity in the HL-60 and HL-60R cell lines. Both cell lines were treated for 48 h with EO at the respective IC_50_ values. Pro-oxidant activity was examined by viable cell count with Trypan blue exclusion test, adding N-acetyl-L-cysteine (NAC), at two concentrations of 1 mM and 2 mM, 1 h before EO. In both cell lines, pretreatment with NAC reduced the cytotoxic activity, confirming that even in these tumor cell lines, the oil acts through a pro-oxidant mechanism (Table 2).

### 2.3. Kinetoplast DNA Decatenation Assay

The presence of hydroxy-methyl-naphthoquinone among the main components of the EO led us to study the ability of the oil to interfere with the activity of topoisomerase II. The effect of EO on the catalytic activity of DNA topoisomerase II was assayed by the kinetoplast DNA (kDNA) decatenation assay. The assay was based upon decatenation of kDNA and because it was specific for type II activity (not type I), it could be carried out with crude cell extracts. Topoisomerase II activity, through introducing dsDNA breaks, can resolve concatenated kinetoplast DNA. The released (decatenated) products are somewhat heterogeneous but are predominantly in the form of nicked open circular minicircles and fully closed circular rings. Both are considered decatenation products. The nuclear extracts obtained from untreated cells, HL-60 and HL-60R (200 ng), were treated or not with EO (at the corresponding IC_50_ values) or with etoposide (60 µg/mL) for 20 min at room temperature and then incubated with kDNA (250 ng) for 30 min at 37 °C. Extracts incubated with kinetoplast DNA converted kDNA from the catenated to the decatenated form; extracts also incubated with EO of *G. rosmarinifolia* inhibited kDNA conversion into decatenated forms, as shown by the reduction in the amount of decatenated DNA substrate (Figure 2). Etoposide was used as a reference drug capable of inhibiting topoisomerase II by stabilizing the cleavage complex. In this case, the positive result is linear DNA, generated by the poisoning effects (linear kDNA). Topo II can only be catalytically inhibited at very high concentrations of etoposide, between 30 and 60 µg/mL. Therefore, in order to visualize this effect as a control, a high concentration of etoposide was used, very strong compared to the IC_50_ of the etoposide in the HL-60 and HL-60R lines.

A very high concentration (100 µg/mL) of EO was also used to verify a possible change in the decatenated products. In both cell lines, etoposide was able to inhibit the formation of decatenated intermediates, but it showed the ability to stabilize the cutting complex by generating linear kDNA. The EO inhibition of topoisomerase II activity was not accompanied by stabilization of a covalent topoisomerase II-DNA intermediate. Thus, EO would seem to act as a catalytic inhibitor rather than as a topoisomerase II poison. 

### 2.4. Plasmid DNA Linearization Assay

To further confirm the mechanism of action of EO as a catalytic inhibitor, a plasmid DNA linearization assay was performed. The cleavage assay experiment was carried out using nuclear extracts from untreated cells (200 ng) subsequently incubated with EO (at the corresponding IC_50_ values) or with etoposide (60 µg/mL) and the supercoiled pBluescript II SK (+) (pSK) plasmid DNA. The linear pSK DNA was identified by comparison with linear pSK DNA produced by the action of the restriction enzyme Pst I acting on a single site on the plasmide. Etoposide was once again used as a positive control to examine whether EO stabilized the cleavable complex. As shown in Figure 3, the addition of etoposide to the experimental mixture induced the formation of linear pSK DNA. On the other hand, the addition of EO to the reaction mixture induced no formation of cleaved linear pSK DNA.

### 2.5. Cell-Cycle Analysis

In order to verify the effect of EO on the cell cycle, a flow cytometry assay with propidium iodide was performed. The two cell lines, HL-60 and HL-60R, were treated with EO (at the respective IC_50_ values), and the distribution of cells within the cell cycle was analyzed after 48 h. Etoposide was used as the reference drug. Figure 4 shows the distribution of HL-60 and HL-60R cells within the cell cycle. In both cell lines, we found that EO induced a G_1_-G_0_ cell cycle arrest, with a reduction in S-phase cells, while etoposide induced arrest in the G_2_-M phase of the cell cycle (Figure 4 and Table 3).

One plausible explanation for the accumulation of cells in G_1_, reflecting the difference in the mechanistic action of EO and etoposide, is that EO prevented DNA replication through debilitating topoisomerase II activity early in the catalytic cycle. Cells may then proceed through the cell cycle until halted by a block in DNA replication. This possibility is consistent with the profound suppression of cells in the S-phase when treated with EO (Figure 3).

### 2.6. Cytotoxic Effects of G. rosmarinifolia EO in Combination with Etoposide and Doxorubicin

Catalytic inhibitors often oppose the effects of topoisomerase II poisons, acting as antagonists, or sometimes, despite the different mechanism of action, they can lead to an enhancement [40,41]. Therefore, in the light of the data obtained, it was finally decided to evaluate the cytotoxic effects of the combination of EO and etoposide. Sub-cytotoxic concentrations were used, and cell growth inhibition was evaluated by MTS assays conducted over 72 h. Table 4 shows the percentages of cell growth inhibition after treatments in the HL-60 and HL-60R cell lines. Cell growth inhibition rates obtained from co-treatments versus predicted percentages showed the potentiation of cell growth inhibition. In addition, we also evaluated the co-treatment of EO and doxorubicin, which was a Topo II inhibitor and induced ROS production. As well as the combination with etoposide, in this case, there was also a potentiation of the cytotoxic effect in both cell lines (Table 4).

## 3. Materials and Methods

### 3.1. Essential Oil

*G. rosmarinifolia* EO was obtained from the same batch as the one used by Poma et al. [7] who hydrodistilled branches with leaves in a Clevenger-type apparatus. They identified the chemical composition by GC analysis using a fused silica Agilent HP 5MS capillary column and a Supelcowax 10 capillary column [7]. To prepare the stock solution for biological assays, 2 mg of essential oil was dissolved in 1 mL of dimethyl sulfoxide (DMSO).

### 3.2. Cell Lines and Culture Conditions

The HL-60 cells were obtained from ATCC^®^ (CCL-240, Rockville, MD, USA), while their variant, HL-60R, were selected for multidrug resistance (MDR) by exposure to gradually increasing concentrations of doxorubicin. The molecular characterization of HL-60R cells was carried out previously [39]. Treatment with doxorubicin modified the phenotype of the HL-60 cell line by inducing the expression of several factors responsible for multidrug resistance. The hTERT RPE-1 cells were kindly provided by Prof. Patrizia Cancemi (Department of Biological, Chemical and Pharmaceutical Science and Technology, University of Palermo, Italy). The HL-60 and HL-60R cells were routinely maintained in Roswell Park Memorial Institute (RPMI) 1640 (HyClone Europe Ltd., Cramlington UK), while hTERT RPE-1 cells were cultured in Dulbecco’s Modified Eagle Medium (DMEM) (HyClone Europe Ltd., Cramlington, UK) supplemented with 10% heat inactivated fetal calf serum, 2 mM L-glutamine, 100 units /mL penicillin and 100 µg/mL streptomycin (all reagents were from HyClone Europe Ltd., Cramlington, UK) in a humidified atmosphere at 37 °C in 5% CO_2_. Cells with a narrow range of passage numbers were used for all experiments. The cultures were routinely tested for Mycoplasma infection.

### 3.3. Cell Growth Inhibition Assays

Exponentially growing cells were suspended at 5 × 10^4^ cells/mL in complete medium, and 100 µL of cell suspension was distributed into each well of 96-well microtiter plates and incubated overnight at 37 °C. At time 0, the medium was replaced with fresh complete medium supplemented with EO and /or etoposide (Sigma Aldrich srl, Milan, Italy) and/or doxorubicin (Sigma Aldrich srl, Milan, Italy) at the indicated concentrations. Following 72 h of treatment, 16 µL of a commercial solution (obtained from Promega Corporation Madison, WI, USA) containing 3-(4,5-dimethylthiazol-2-yl)-5-(3-carboxymethoxyphenyl)-2-(4-sulphophenyl)-2H-tetrazolium (MTS) and phenazine ethosulfate was added. The plates were incubated for 4 h in a humidified atmosphere at 37 °C in 5% CO_2_. The bioreduction in the MTS dye was assessed by measuring the absorbance of each well at 490 nm. The inhibition rate on cell proliferation (mean±SE) was calculated as (1-(A_490_ treated/A_490_ control)) × 100%. The IC_50_ value was determined from the results of at least three independent tests. The resistance factor (RF) to each drug was calculated as the ratio of the IC_50_ value of resistant cells to that of parental cells.

### 3.4. Pro-Oxidant Activity

To determine cell pro-oxidant activity, HL-60 and HL-60R cells (1 × 10^5^) were treated for 48 h with *G. rosmarinifolia* EO at a concentration equal to IC_50_ in the two cell lines. Pro-oxidant activity was examined by viable cell count with Trypan blue exclusion test, adding N-acetyl-L-cysteine (NAC) (Sigma Aldrich srl, Milan, Italy), an antioxidant molecule, at two concentrations of 1 mM and 2 mM, 1 h before EO. Data were expressed as mean ± standard error (SE) of at least three different experiments performed in duplicate.

### 3.5. Preparation of Nuclear Extracts

Exponentially growing HL-60 and HL-60R cells were collected, washed once with PBS, suspended in 3 mL of cold TEMP buffer (10 mM Tris–HCl, pH 7.5; 1 mM EDTA; 4 mM MgCl_2_; 0.5 mM PMSF) and incubated in ice for 15 min. The lysed cells were centrifuged at 1500 g for 10 min. The pelleted nuclei were washed twice with TEMP buffer, centrifuged again as described above, resuspended in 2 volumes of cold TEP buffer (same as TEMP but lacking MgCl_2_) plus an equal volume of 1 M NaCl and incubated in ice for 60 min. The samples were then centrifuged at 15,000 g for 20 min. Total protein amount was measured using the Bradford assay.

### 3.6. Kinetoplast DNA Decatenation Assay

Kinetoplast DNA (kDNA; TopoGEN, Port Orange, FL, USA) decatenation assays were performed by utilizing nuclear extracts (200 ng) from untreated HL-60 and HL-60R cells. According to the manufacturer’s instructions, supercoiled plasmid DNA (kDNA, 200 ng) was incubated in 20 µL of reaction buffer [50 mM Tris–HCl (pH 8.0), 120 mM KCl, 10 mM MgCl_2_, 0.5 mM ATP, and 0.5 mM DTT]. Reactions were carried out at 37 °C for 30 min and then halted by the addition of 4 µL of stop buffer (5% sarkosyl, 0.0025% bromophenol blue, 25% glycerol). Pre-incubation (20 min) of extracts and EO (36.7 µg/mL or 37 µg/mL and 100 µg/mL) or etoposide (60 µg/mL) was carried out at room temperature and the reaction initiated by the addition of plasmid and transfer to 37 °C. Samples were separated on a 1% agarose gel with ethidium bromide 0.5 µg/mL for 30 min. DNA bands were visualized by ultraviolet light. Double-stranded DNA cleavage was monitored by the conversion of supercoiled plasmid DNA to decatenation molecules. Inhibition of topoisomerase was evidenced by the reduction in intensity of decatenated kDNA products. Etoposide was used as a positive control (inhibitor of topoisomerase-II capable of stabilizing the cleavage complex).

### 3.7. Plasmid DNA Linearization Assay

DNA cleavage assays using nuclear extracts (200 ng) from untreated cells were performed in 20 µL of reaction mixture containing 150 ng of supercoiled pBluescript II SK (+) plasmid DNA, 0.5 mM ATP in assay buffer [10 mM Tris–HCl, 50 mM KCl, 50 mM NaCl, 0.1 mM EDTA, 5 mM MgCl_2_, 2.5% (*v*/*v*) glycerol, pH 8.0], EO (at the corresponding IC_50_ values) or etoposide (60 µg/mL). The order of addition was assay buffer, DNA, EO or etoposide, and then, nuclear extracts. The reaction mixture was incubated at 37 °C for 30 min, quenched with 1% (*v*/*v*) SDS/25 mM Na_2_EDTA and then treated with 0.25 mg/mL proteinase K (Invitrogen Life Technologies, Carlsbad, CA, USA) at 55 °C for 60 min. The samples were separated by electrophoresis on a 1% TAE ethidium bromide agarose gel, and the linear pBluescript II SK (+) DNA was identified by comparison with linear pBluescript II SK (+) DNA produced by the action of the restriction enzyme Pst I (New England BioLabs, Beverly, MA, USA) acting on a single site on pBluescript II SK (+).

### 3.8. Cell-Cycle Analysis

To determine cell-cycle distribution, HL-60 and HL-60R cells (1 × 10^5^) were treated for 48 h with *G. rosmarinifolia* EO or etoposide (used at the respective IC_50_ for the two lines). After treatment, cells were collected and washed twice with ice-cold PBS and then resuspended at 1 × 10^6^ /mL in a hypotonic fluorochrome solution containing propidium iodide (PI) 50 μg/mL and RNase (10 mg/mL) in 0.1% sodium citrate plus 0.03% (*v*/*v*) Nonidet P-40. After 45 min at room temperature (in the dark) of incubation in this solution, the samples were filtered through a nylon cloth, 40 μm mesh, and samples were analyzed using a FACSCanto instrument (Becton Dickinson, Montain View, CA, USA). The data were analyzed with BD FACSDiva software v.6.1.2. (Becton Dickinson). Cell distribution was determined by evaluating the percentage of events accumulated in the different phases of the cycle.

### 3.9. Statistical Analysis

The results are expressed as the average of three repetitions ± standard error. Statistical analysis was carried out with the analysis of variance (one-way ANOVA) followed by Tukey’s test using Statistics ver. 12 (StatSoft Inc., Oklahoma City, USA, 1984–2014).

## 4. Conclusions

*G. rosmarinifolia* EO caused cytotoxicity in terms of cell growth inhibition and cell-cycle variation both in the HL-60 and HL-60R cell lines. EO was not subject to cell resistance through P-gp and sensitized the MDR cell line to the cytotoxic effects of P-gp substrates, doxorubicin and etoposide. The antitumor activity of EO on the HL-60 cell line and its MDR variant depends on a pro-oxidant action.

*G. rosmarinifolia* EO acts as a modulator of Topo II through a different mechanism to etoposide; EO acts as a catalytic inhibitor rather than a Topo II poison. Unlike etoposide, which induces arrest in the G_2_-M phase of the cell cycle, EO induces a cell-cycle arrest in the G_0_-G_1_ phase of the cell cycle. The reduction in the S-phase cells by EO supports the hypothesis that it works by preventing Topo II from functioning properly at an early stage.

In light of these considerations, *G. rosmarinifolia* EO could be used as an adjuvant in chemotherapy regimens.

## Figures and Tables

**Figure 1 molecules-27-04203-f001:**
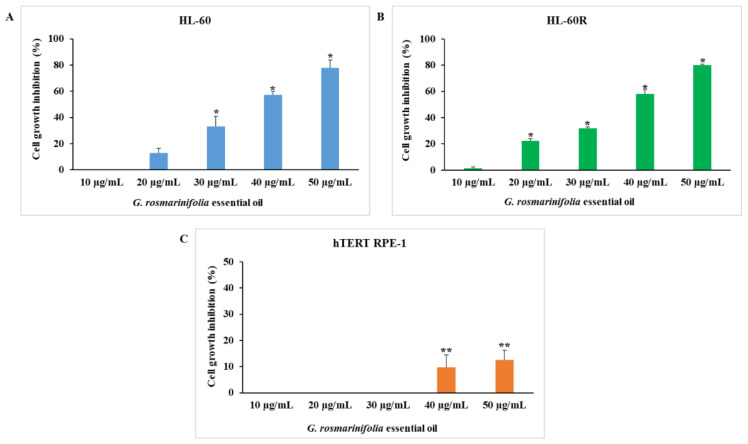
Cytotoxic activity of *G. rosmarinifolia* EO. Cell viability was assessed by MTS after 72 h of treatment at different concentrations. (**A**) HL-60 cell line; (**B**) HL-60R cell line; (**C**) hTERT RPE-1 cell line. Data are expressed as mean ± standard error (SE) of at least three different experiments performed in triplicate. Differences when treatments are compared to the controls: * *p* < 0.001; ** *p* < 0.05 (one-way ANOVA followed by Tukey’s test).

**Figure 2 molecules-27-04203-f002:**
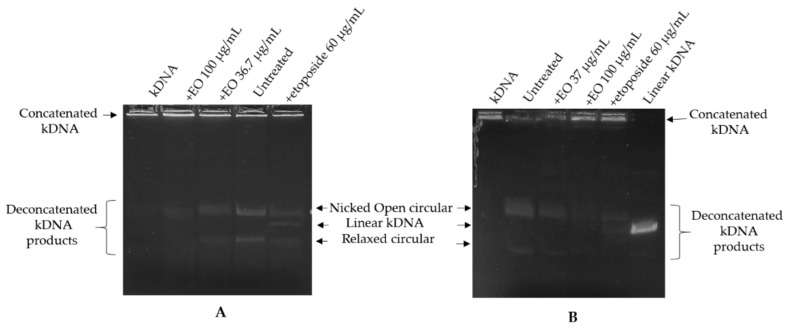
Inhibition of topoisomerase II activity in HL-60 (**A**) and HL-60R (**B**) cell nuclear extracts by *G. rosmarinifolia* essential oil. Nuclear extracts (200 ng) were treated or not with essential oil, or with etoposide and incubated with kDNA. The panels show a representative experiment of three independent experiments. The decatenated products shown contain open circular (upper band) and covalently closed circular (relaxed) minicircle DNA. Linear kDNA migrates between the affected and relaxed species.

**Figure 3 molecules-27-04203-f003:**
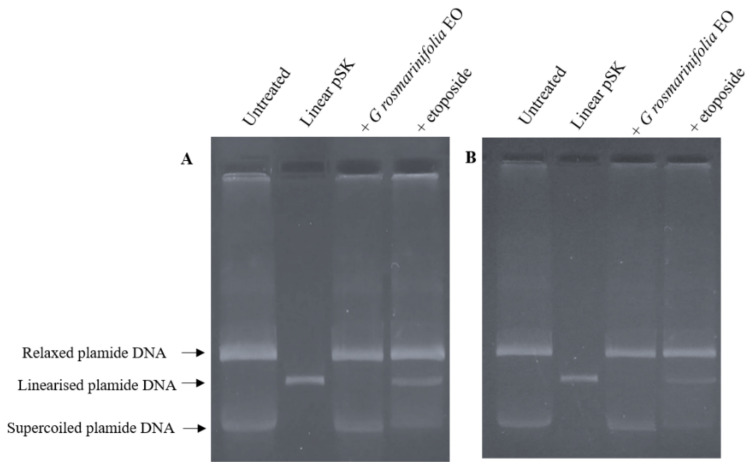
Plasmid DNA linearization assay. Nuclear extracts from untreated cells, HL-60 (**A**) and HL-60R (**B**), were incubated with EO (at the corresponding IC_50_ values) or with etoposide (60 µg/mL) and the supercoiled pSK plasmid DNA. Etoposide was used as a control capable of stabilizing the cleavable complex. Linear pSK DNA produced by the action of the restriction enzyme Pst I acting on a single site on the plasmide. The panels show a representative experiment of three independent experiments.

**Figure 4 molecules-27-04203-f004:**
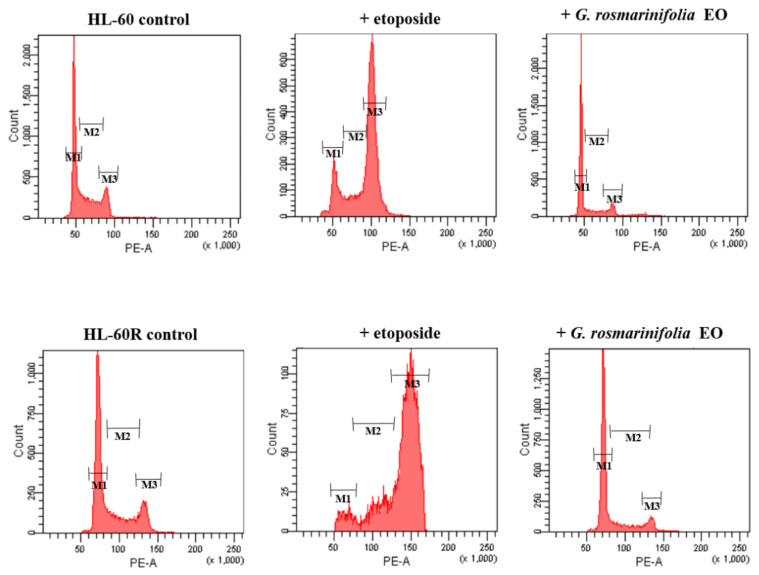
Cell-cycle analysis in the HL-60 and HL-60R cell lines. Cells were treated for 48 h with IC_50_ of *G. rosmarinifolia* essential oil or etoposide. EO induced cell-cycle arrest in the G_0_-G_1_ phase of the cell cycle, with a reduction in cells in the S-phase. The panel shows a representative experiment of three independent experiments.

**Table 1 molecules-27-04203-t001:** IC_50_ values and resistance factors of *G. rosmarinifolia* EO and etoposide in the cell lines.

		IC_50_ (Mean ± SE) µg/mL		Resistance Factor (RF)
	**HL-60**	**HL-60R**	**hTERT RPE-1**	
*G. rosmarinifolia* EO	36.75 ± 0.2	37.0 ± 0.7	>50.0	1
etoposide	0.05 ± 2.9	7.4 ± 0.6	NT	148.0

**Table 2 molecules-27-04203-t002:** Results of cell counting analysis in HL-60 and HL-60R cell lines following treatment with antioxidant N-acetyl-L-cysteine (NAC) at two concentrations, 1 mM and 2 mM, before exposure to EO at the corresponding IC_50_.

Cell Lines and Treatments	Cell Viability (%)
**HL-60**	
NAC 1 mM	100.0± 0.3
NAC 2 mM	81.5 ± 6.4
essential oil of *G. rosmarinifolia* 36.7 µg/mL	49.5 ± 5.3 *
NAC 1 mM + essential oil of *G. rosmarinifolia* 36.7 µg/mL	81.5 ± 3.9
NAC 2 mM + essential oil of *G. rosmarinifolia* 36.7 µg/mL	80.0 ± 2.8
**HL-60R**	
NAC 1 mM	100.0 ± 0.3
NAC 2 mM	100.0 ± 2.1
essential oil of *G. rosmarinifolia* 37 µg/mL	47.5 ± 4.3 *
NAC 1 mM + essential oil of *G. rosmarinifolia* 37 µg/mL	76.5 ± 0.3 *
NAC 2 mM + essential oil of *G. rosmarinifolia* 37 µg/mL	97.0 ± 1.4

Data are expressed as mean ± standard error (SE). Differences when treatments are compared to the control: * *p* < 0.05 (one-way ANOVA followed by Tukey’s test).

**Table 3 molecules-27-04203-t003:** Cell-cycle changes induced by *G. rosmarinifolia* EO in HL-60 and HL-60R cells.

	G_0_/G_1_	S	G_2_/M
**HL-60**	45.0 ± 0.47	35.0 ± 0.94	18.3 ± 1.0
+ etoposide	8.1± 1.65 ***	19.1 ± 0.41 ***	70.1 ± 0.52 ***
+ *G. rosmarinifolia* EO	69.0 ±1.88 ***	14.8 ± 0.88 ***	12.6 ± 0.17 *
**HL-60R**	60.4 ± 0.67	22.4 ± 1.14	15.9 ± 0.52
+ etoposide	7.4 ± 0.29 ***	12.3 ± 0.61 ***	79.5 ± 0.71 ***
+ *G. rosmarinifolia* EO	73.6 ± 0.76 ***	13.0 ± 0.47 **	13.1 ± 0.52

Cells were treated for 48 h with the agents at the corresponding IC_50_ values, and their distribution in the phases of the cell cycles was assessed through a flow cytometry analysis of their DNA stained with propidium iodide. Data are the mean ± S.E. of three separate experiments. * *p* < 0.01, ** *p* < 0.005 and *** *p* < 0.001 versus control (one-way ANOVA followed by Tukey’s test).

**Table 4 molecules-27-04203-t004:** Cytotoxic effects of essential oil in combination with etoposide and doxorubicin in HL-60 and HL-60R cells.

Treatments	Cell Growth Inhibition, %	Expected(%)
**HL-60**		
essential oil 20 µg/mL	13.0 ± 2.1	
essential oil 30 µg/mL	31.2 ± 5.6 *	
etoposide 0.005 µg/mL	15.0 ± 0.9	
etoposide 0.01 µg/mL	22.0 ± 0.9 *	
doxorubicin 0.002 µg/mL	8.0 ± 2.1	
doxorubicin 0.005 µg/mL	21.0 ± 0.3	
essential oil 20 µg/mL + etoposide 0.005µg/mL	38.0 ± 0.7 **	26.0 ± 0.5 * ^b^
essential oil 20 µg/mL + etoposide 0.01 µg/mL	55.0 ± 0.4 **	32.0 ± 0.2 ** ^a^
essential oil 30 µg/mL + etoposide 0.005 µg/mL	54.0 ± 0.9 **	43.0 ± 0.2 ** ^b^
essential oil 30 µg/mL + etoposide 0.01 µg/mL	60.0 ± 1.2 **	46.0 ± 1.4 ** ^a^
essential oil 20 µg/mL + doxorubicin 0.002 µg/mL	35.0 ± 0.1 **	20.0 ± 1.4 ^a^
essential oil 20 µg/mL + doxorubicin 0.005 µg/mL	80.0 ± 1.3 ***	32.0 ± 1.4 * ^a^
essential oil 30 µg/mL + doxorubicin 0.002 µg/mL	55.0 ± 1.2 **	39.0 ± 0.7 ** ^a^
essential oil 30 µg/mL + doxorubicin 0.005 µg/mL	83.0 ± 1.5 ***	47.0 ± 2.1 ** ^a^
**HL-60R**		
essential oil 20 µg/mL	22.3 ± 2.1 *	
essential oil 30 µg/mL	32.0 ± 1.6 *	
etoposide 2.5 µg/mL	11.0 ± 0.9	
etoposide 5 µg/mL	33.0 ± 0.7 *	
doxorubicin 1 µg/mL	12.0 ± 2.6	
doxorubicin 5 µg/mL	32.0 ± 1.4 *	
essential oil 20 µg/mL + etoposide 2.5µg/mL	42.0 ± 0.7 **	31.0 ± 0.5 ** ^b^
essential oil 20 µg/mL + etoposide 5 µg/mL	65.0 ± 0.7 **	48.0 ± 0.7 ** ^a^
essential oil 30 µg/mL + etoposide 2.5 µg/mL	53.0 ± 0.6 **	40.0 ± 0.4 ** ^b^
essential oil 30 µg/mL + etoposide 5 µg/mL	77.0 ± 0.5 ***	55.0 ± 0.8 ** ^a^
essential oil 20 µg/mL + doxorubicin 1 µg/mL	44.0 ± 2.2 *	31.0 ± 0.6 * ^a^
essential oil 20 µg/mL + doxorubicin 5 µg/mL	59.0 ± 0.7 **	48.0 ± 0.9 ** ^b^
essential oil 30 µg/mL + doxorubicin 1 µg/mL	56.0 ± 0.5 **	40.0 ± 2.3 ** ^a^
essential oil 30 µg/mL + doxorubicin 5 µg/mL	67.0 ± 0.9 **	53.0 ± 1.7 ** ^a^

Data are expressed as mean ± standard error (SE). *** *p* < 0.001, ** *p* < 0.005, * *p* < 0.01 versus controls; ^a^
*p* < 0.001 and ^b^
*p* < 0.01 expected versus observed (one-way ANOVA followed by Tukey’s test).

## Data Availability

Data are contained within the article.

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
