# Peer review of "Antitumor Effect of Glandora rosmarinifolia (Boraginaceae) Essential Oil through Inhibition of the Activity of the Topo II Enzyme in Acute Myeloid Leukemia"

_molecules, 2022, doi:10.3390/molecules27134203_

Round 1
Reviewer 1 Report
The manuscript "Antitumor effect of Glandora rosmarinifolia (Boraginaceae) essential oil by inhibiting the activity of the Topo II enzyme in acute myeloid leukemia" is very interesting and high quality paper. However, there is one deficiency. In Material and method part, there are no data about plant material, essential oil extraction, chemical composition of used EO...
Author Response
Reviewer 1
The manuscript "Antitumor effect of Glandora rosmarinifolia (Boraginaceae) essential oil by inhibiting the activity of the Topo II enzyme in acute myeloid leukemia" is very interesting and high quality paper. However, there is one deficiency. In Material and method part, there are no data about plant material, essential oil extraction, chemical composition of used EO...
We thank the reviewer for his/her comment. We have inserted a paragraph (3.1) with information and data on the essential oil in Materials and Methods section.
Reviewer 2 Report
I recommend a few modification.
Please describe the abbreviations when you use them for the first time
Rephrase between row 203 and 209 its hard to read!
The paragraph 2.2 need to rephrase this part.
I think that you need three short conclusions!
My recommendation is to focus on 3 short conclusion.
Double check the English.
Author Response
I recommend a few modification.
Please describe the abbreviations when you use them for the first time
We thank the reviewer for his/her suggestion. We have described the abbreviations in the text.
Rephrase between row 203 and 209 its hard to read!
We have rephrased row 203 and 209.
The paragraph 2.2 need to rephrase this part.
We have rephrased paragraph 2.2.
I think that you need three short conclusions!
My recommendation is to focus on 3 short conclusion.
Double check the English.
We double-checked the English and rephrased the conclusions as suggested.
Reviewer 3 Report
The manuscript titled Antitumor effect of Glandora rosmarinifolia (Boraginaceae) es- 2 sential oil by inhibiting the activity of the Topo II enzyme in 3 acute myeloid leukemia by authors et al. highlight the antitumor activity of Essential oil(EO) on the HL-60 cell line and its MDR variant with a peculiar mechanism as a Topo II modulator, which demonstrated the G. rosmarinifolia EO a potential anticancer drug candidate. In general, this paper seems to be quite interesting and I would like to recommend the acceptance of this work provided that authors can well address the following questions.
1.The GC-MS assay of G. rosmarinifolia EO was suggested.
2. The dissolution method of G. rosmarinifolia EO used in cell experiment should be described.
Author Response
The manuscript titled Antitumor effect of Glandora rosmarinifolia (Boraginaceae) essential oil by inhibiting the activity of the Topo II enzyme in 3 acute myeloid leukemia by authors et al. highlight the antitumor activity of Essential oil(EO) on the HL-60 cell line and its MDR variant with a peculiar mechanism as a Topo II modulator, which demonstrated the G. rosmarinifolia EO a potential anticancer drug candidate. In general, this paper seems to be quite interesting and I would like to recommend the acceptance of this work provided that authors can well address the following questions.
1.The GC-MS assay of G. rosmarinifolia EO was suggested.
We thank the reviewer for his/her comment. We have included a paragraph in Materials and Methods section where the previously performed G. rosmarinifolia EO GC-MS assay is described (3.1).
- The dissolution method of G. rosmarinifolia EO used in cell experiment should be described.
We thank the reviewer for his/her suggestion. We have included the method of dissolving the G. rosmarinifolia EO in 3.1 of Materials and Methods section.